

# Occurrence of structural aluminium (Al) in marine diatom biological silica: Visible evidence from microscopic analysis

Qian Tian[1,2,3], Dong Liu[1,2,4*], Peng Yuan[1,2], Mengyuan Li[1,2,3], Weifeng Yang[4], Jieyu Zhou[1,2,3], Huihuang Wei[1,2,3], Junming Zhou[1,2,3], & Haozhe Guo[1,2,3]

[1] *CAS Key Laboratory of Mineralogy and Metallogeny/Guangdong Provincial Key Laboratory of Mineral Physics and Materials, Guangzhou Institute of Geochemistry, Institutions of Earth Science, Chinese Academy of Sciences, Guangzhou 510640, China*
[2] *CAS Center for Excellence in Deep Earth Science, Guangzhou, 510640, China*
[3] *University of Chinese Academy of Sciences, Beijing 100049, China*
[4] *State Key Laboratory of Marine Environmental Science, Xiamen University, Xiamen, 361102, China*

*Correspondence to*: Dr. Dong Liu (liudong@gig.ac.cn)
Current addresses: Guangzhou Institute of Geochemistry, Chinese Academy of Sciences Wushan, Guangzhou 510640, China


**Abstract.**

The global marine biogeochemical cycle of aluminum (Al) is believed to be driven by marine diatoms, due to the uptake of dissolved Al (DAl) by living diatoms from surface seawater. The occurrence of Al in diatom biogenic silica (BSi) can inhibit the dissolution of BSi, thus benefiting the effects of the ballast role of diatoms in the biological pump and forming a coupled 20 Si-Al biogeochemical cycle. However, the occurrence mechanism of Al in marine diatoms is still unclear. In particular, whether or not Al is incorporated into the structure of BSi of living diatoms is unrevealed, resulting in difficulties in understanding the biogeochemical behaviors of Al. In this study, Thalassiosira weissflogii, a widely distributed marine diatom in marginal seas, was selected as the model to evaluate the occurrence of structural Al in BSi based on culturing experiments with the addition of DAl. The structural Al in BSi was detected by combining focused ion beam (FIB) scanning 25 electron microscopy and energy dispersive X-ray spectroscopy (EDS) mapping analysis. Direct evidence of structural Al in living BSi was obtained for the first time. The distribution and content of this Al were revealed by the EDS-mapping analysis. The structural Al in the BSi exhibited a homogeneous distribution, and the average Al/Si atomic ratio obtained through the FIB-EDS mapping analysis was 0.011. The effects of structural Al on BSi dissolution-inhibition are discussed based on the content of this Al. The fundamental results indicate the significant contribution of marine diatoms to the 30 biogeochemical migration of marine Al.

## 1 Introduction

Aluminum (Al) is the most abundant metal element in the Earth's crust, and it mainly occurs in Al-bearing minerals such as aluminosilicate (*Taylor*,1964). The weathering of such aluminosilicates results in a huge carbon (C) sink in rocks and soils





due to the transformation of $CO_2$ to $HCO_3^-$ based on the reaction with such Al-bearing minerals (*Mackenzie and Kump*, 1995; *Wallmann et al.*, 2008). Therefore, Al is involved in the terrestrial C cycle (*Stallard*, 1998). Despite the high-Al content of the terrestrial system, the concentration of dissolved Al (DAl) in natural water is very low (from < tens of nM in oceans to µM in freshwater) (*Hydes*, 1977; *Measures and Edmond*, 1990; *Chou and Wollast*, 1997; *Menzel et al.*, 2020), due to the low dissolution of Al-bearing minerals; and thus, oceanic DAl is often used as a tracer for the transportation of terrestrial

substance and the mixing of different water masses (*Measures and Vink*, 1999; 2000; *Han et al.*, 2008; *Schlitzer et al.*, 2018). Due to its lack of an essential biological role (even toxicity) (*Xie et al.*, 2015; *Gillmore et al.*, 2016), the terrestrial cycling of Al is believed to be primarily controlled by inorganic processes. However, marine diatoms are found to take up and incorporate Al into their cells (*Q Liu et al., 2019*). Diatoms can eliminate plenty of DAl from surface water when they bloom (*Hall et al.*, 1999; *J-L Ren et al.*, 2011), and they export Al into the sedimentary layers through the sedimentation of post-

mortem diatoms, forming a coupled Si-Al biogeochemical cycle in the oceans (*van Hulten et al.*, 2013; 2014).

Diatoms are a type of widely distributed single-cell algae in the oceans, and they account for up to ~40% of the oceanic primary production (*Armbrust*, 2009). Through sedimentation of dead diatoms, huge amounts of organic carbon (OC) are exported from the surface ocean to the deep ocean (*Nelson et al.*, 1996; *Riebesell*, 2000; *Leblanc et al.*, 2018) controlling the oceanic C cycle and forming an important part of the biological carbon pump (*Smetacek*, 1999; *Tréguer and Pondaven*, 2000;

*Ragueneau et al.*, 2006).

Compared with other marine phytoplankton, diatoms are more effective at carbon sequestration over longer timescale due to their frustules, which are composed of biogenic silica (BSi) with a high-mechanical stability. Since it acts as ballast (*De La Rocha et al.*, 2008; *Honda and Watanabe*, 2010), BSi carries the OC to deeper oceans and deposits on the seafloor, forming a coupled Si-C cycle.

Considering the key role of diatoms in carbon sequestration, the Al in diatom cells is also involved in the C cycle in the oceans, and thus, Al participates in the global C geochemical cycle through inorganic and biological processes (*Gehlen et al.*, 2003; *H Ren et al.*, 2013). Although preliminary studies have been carried out to investigate the Al in freshwater (*D Liu et al.*, 2019) and marine diatoms, the occurrence of Al in marine diatoms is far away unclear, which results in difficulties in understanding the abovementioned diatom-derived Al cycle.

It has been proposed that Al is incorporated into the organic components and BSi of diatoms. For the former, the occurrence mechanism and distribution have been well described (*Q Liu et al.*, 2019). However, for the latter, whether Al occurs in the structure of BSi is ambiguous. Many researches have tried to prove the existence of structural Al based on the detection of BSi sourced from living marine diatoms and diatom fossils by removing the organic component of the diatoms and the Al-bearing matter on the surface, respectively (*Gehlen et al.*, 2002; *Koning et al.*, 2007; *Machill et al.*, 2013). However, Al-

bearing tiny mineral particles can be hardly excluded from sedimentary BSi even with the greatest of care, e.g., the acid washing reported by Gelhen (*Gehlen et al.*, 2002). Moreover, the high activity of fresh BSi results in the easy adsorption of DAl from Al-contained solutions during the removal of the Al-bearing organic components of diatoms, which interferes with the evaluation of the structural Al (*Moran and Moore*, 1988; *Koning et al.*, 2007). Due to these challenges, the occurrence of





Al in the BSi structure still unclear, although possible contents and coordinated states have been proposed. The reason that
so much attention has been paid to the structural Al in BSi is that structural Al has a dissolution-inhibition effect on BSi.
About 25% decrease in the solubility of BSi was speculated when Al substitutes for 1 out of every 70 Si atoms (*Dixit et al.*,
2001),and thus, structural Al is believed to be one of the key factors that influence the migration of BSi from the surface
ocean to pelagic sediments. Therefore, the occurrence of Al in BSi is indefinite, which severely limits our understanding of
the composition, structure, and water stability of BSi.

In this study, a widely distributed marine diatom in marginal seas, *Thalassiosira weissflogii* (*T. weissflogii*)
(*Panagiotopoulos et al.*, 2020), was selected as the model; and culture experiments were performed with the addition of high
concentrations of DAl. The direct detection of the internal surface of BSi was conducted using a focused ion beam (FIB)
pretreatment, which can remove the external surface, leading to purification of BSi. Energy dispersive X-ray spectroscopy
(EDS) mapping analysis was used to determine the content and distribution of the structural Al. Finally, the coupled Si-Al
cycle controlled by marine diatoms was discussed.



## 2 Materials and Methods

### 2.1 Cultivation of the diatoms

*T. weissflogii*, purchased from the Shanghai Guangyu Biological Technology Co., Ltd (China), was used as the model

diatom in this study to investigate Al occurrence. The species of the marine diatom samples was identified by determining

their rbcL gene sequences (*Couradeau et al.*, 2012). The obtained phylogram clearly showed that the diatoms obtained was *T.*

*weissflogii* (Fig. 1).

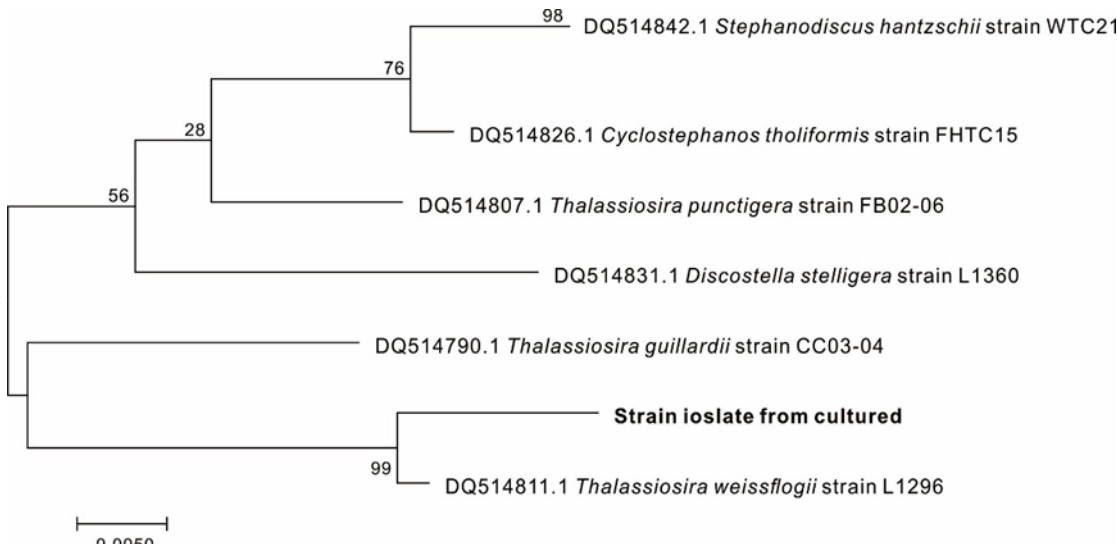

**Fig. 1. Phylogenetic trees of diatom strains isolated from the culturing diatoms. The scale bar indicates the number of substitutions**
**per site for a unit branch length.**

The diatoms were cultured in artificial seawater supplemented with f/2 medium at 25°C under a 12/12 light/dark cycle at an

intensity of 100 mol photons $m^{-2}s^{-2}$ for 20 days. The composition of artificial seawater and f/2 culture medium are presented

in Table. S1 (see Supporting Information). Deionized water was used to prepare the artificial seawater and f/2 medium. Cell

of the diatoms in the culture media with/without the addition of Al was counted using flow cytometry (BioMarine-IA 1000,

Countstar Company, Shanghai, China).

Trace-metal clean processes were used to eliminate the interference from other elements. Before the cultivation experiment,

all of the bottles and tubes were soaked in 10% hydrochloric acid for 24 h, sterilized at 120°C for 30 min, and washed

repeatedly with deionized water (*Q Liu et al.*, 2019).

$AlCl_3$ (i.e., the Al source) was added to the culture medium with an initial concentration of 2.0 μM in the medium. The

control experiment was also simultaneously performed without adding $AlCl_3$. The pH value of the diatom culture medium

was adjusted to 8.0 using NaOH solution. All of the chemical reagents were allowed to stand overnight to reach chemical

equilibrium before use.





## 2.2 Collection of diatoms and extraction of diatom frustules

After 14 days of culturing, the diatoms were collected through high-speed centrifugation at 11000 rpm. The obtained diatoms (Fig. 2) were rinsed with deionized water three times to remove any impurities adsorbed on the surface. The solid obtained was freeze-dried after centrifugation. Two pre-treatment procedures were used to remove any Al adsorbed on the surface of the diatoms and their organic components in order to obtain pure BSi: 1) immersion in 0.05 M EDTA for 12 h followed by washing three times; and 2) immersion in 30% hydrogen peroxide solution for 48 h followed by washing five times. The removal of the organic components was confirmed using a Vario EL III elemental analyzer with a detection limit of <40 ppm. Approximately 2.00 mg of sample was placed in a $6 \times 12$ mm tin boat and was oxidized in a combustion tube in the presence of oxygen at high temperature ($\geq$1150°C).

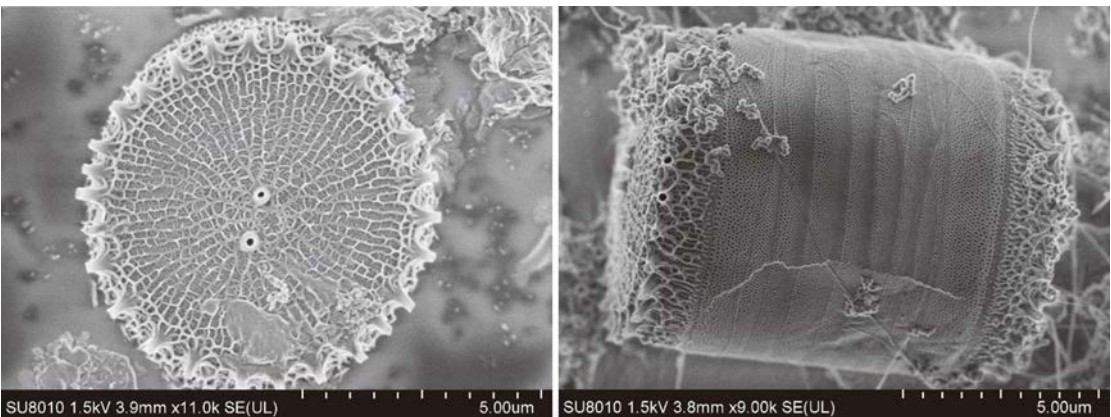

**Fig. 2. Scanning electron microscopy (SEM) of *T. weissflogii***

## 2.3 Characterization and Data analysis

2.3.1 Focused ion beam (FIB) treatment and characterization methods

The FIB thinning was carried out using FIB scanning electron microscopy (SEM; FEI Helios Nano Lab 450S) equipped with a flip stage, an in situ scanning transmission electron microscope (STEM) detector, a Tomahawk ion column, and a multichannel gas injection system. For the FIB milling conducted to obtain a thin BSi slice, a 5 kV focused gallium ion (Ga$^+$) beam with a beam current of 40 pA was used, and the total thinning time was 2 min. Then, the obtained BSi slices were fixed on the edge of the FIB half grid through induced platinum (Pt) deposition. The in situ field emission SEM (FE-SEM) observation of the BSi slice was performed with an accelerating voltage of 30 kV and a current of 24–9300 pA. The elemental distribution of the sliced BSi was obtained using an energy-dispersive X-ray spectrometer attached to a transmission electron microscope (FEI Talos F200 TEM/EDS microscope) with a voltage of 200 kV.

2.3.2 Energy dispersive spectroscopy (EDS) spot analysis

In previous studies, the change in the Al concentrations of the culture medium was used to evaluate the Al uptake of diatoms. However, some of the Al form precipitated due to changes in the chemical environment, rather than being involved in the





biological processes of diatoms. To avoid the disturbance of such Al, which does not incorporate into the cells of the diatoms, the Al/Si atomic ratios of the cultured diatoms and their BSi were determined using EDS spot analysis and a scanning electron microscope (SU8010, UltraPore-300, PDP-200). A total of 353 spots from 35 whole diatoms and the BSi of the *T. weissflogii* were detected, and the Al/Si atomic ratios were obtained at a voltage of 15 kV and a current of 20 μA. Carbon

135 coating was performed before the SEM-EDS analysis.

The data for the specific growth rate and EDS spot analysis were obtained on the basis of the mean ± standard deviation (S.D.; $n \geq 3$). One-way analysis of variance (ANOVA) was used to determine the significant differences ($p < 0.05$) between the treatments (*Shi et al.*, 2015).

140 **3 Result**

**3.1 Effects of Al on the growth of diatoms**

To investigate the influence of Al on the growth of *T. weissflogii*, the cell density was measured during the diatom cultivation processes. The stable growth period of the diatoms cultured with and without the addition of Al began on the third day. The growth trends of the diatom cells were similar in the diatom culture mediums with and without the addition of

145 Al (Fig. 3). The final concentration of the diatom cells exhibited a very slight difference, with average values of $1.96 \times 10^5$ and $1.91 \times 10^5$ cells/mL for the Al-free diatom culture medium and the Al-added medium, respectively. The results indicate that the presence of Al did not significantly affect the cell yields of *T. weissflogii* during the 14-day culture period.

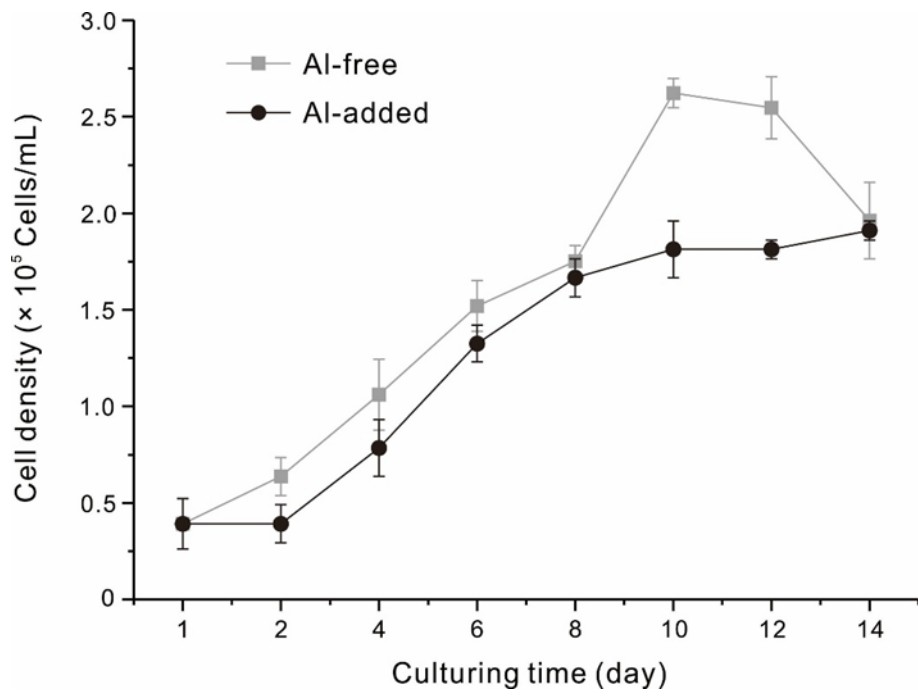



**Fig. 3. The cell densities of the diatoms cultured in culture mediums with (Al-added) and without (Al-free) the addition of Al during the 14-day culturing period.**

### 3.2 Identification of structural Al in BSi based on FIB-EDS

FIB milling can remove the external surface of the diatom and impurities adsorbed onto it, allowing for the detection of the inner structure of the BSi. Thus, the subsequent EDS detection obtains the structural element distributions of the BSi. In this case, the combined FIB–EDS analyses avoided the disturbance of the non-structural elements. The *T. weissflogii* BSi sample was selected (Fig. 4a and the inset) and milled to a thin slice using FIB (Figs. 4b and c). Two inner areas of the BSi slice were observed (blue rectangles in Fig. 4d), and the Al distribution was detected in these two domains, revealing a homogeneous distribution of Al accompanied by Si in the BSi structure (Figs. 4e and g).

**Fig. 4. (a and the inset) FESEM image of a *T. weissflogii* BSi; (b and c) FESEM images of a series of FIB thinning products of the BSi during the ion sputtering experiments; (d) HRTEM bright field (BF) image of the selected domain in the rectangle in (c); (e and g) Al and (f and h) Si distributions in the domains in (d) obtained using EDS mapping analysis.**

This is direct evidence of the presence of Al in the internal surface of BSi sourced from living marine diatoms, demonstrating the occurrence of structural Al in the diatomaceous biological framework of marine diatoms. The Al is incorporated into the BSi based on the biological behaviors of marine diatoms and is used to build diatom BSi. Based on the EDS mapping analysis, the average Al/Si atomic ratio is 0.011.

### 3.3 Quantity measurement of the Al in the diatoms and their BSi using EDS spot analysis





More than 300 spots on diatoms and their BSi were analyzed using EDS, and the average Al/Si atomic ratios were 0.086 and 0.030 for the entire diatoms and their BSi, respectively. The Al/Si value of the BSi is lower than that of the entire diatoms (Fig. 5), because Al not only occurs in BSi but also in the organic components of the diatoms. The Al content of the organic

component is nearly 3-fold higher than that of the BSi. Moreover, the average Al/Si atomic ratio of the BSi (0.030) obtained through EDS spot analysis is higher than that obtained through FIB-EDS mapping (0.010).

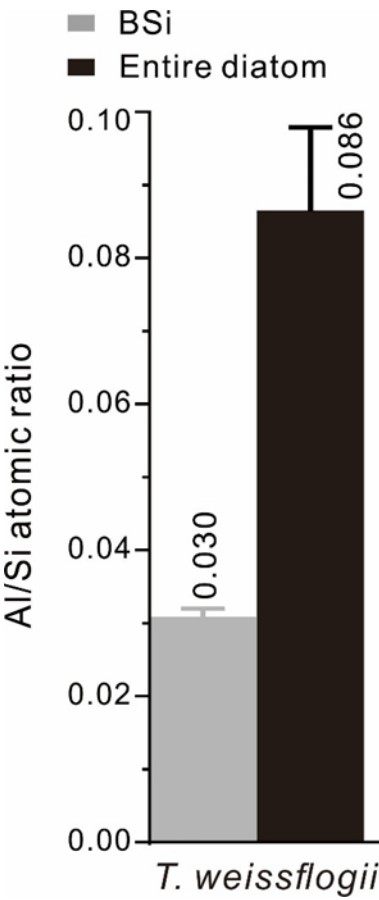

**Fig. 5 Al/Si atomic ratios of the *T. weissflogii* diatom cells (the black column) cultured in media with the addition of Al and of their**

**BSi (the grey column). The EDS spot analysis used to determine the Al/Si atomic ratios was based on 353 spots on 35 diatom cells and their BSi samples. The error bars were obtained through mean ± standard deviation analysis of all of the Al/Si atomic ratios**

## 4 Discussion

The scavenging of Al from seawater by diatoms has been widely observed during diatom blooms (*Hall et al.*, 1999; *J-L Ren*

*et al.*, 2011). This unique biological behavior of Al uptake has attracted a great deal of attention, and a corresponding biogeochemical Al cycle driven by diatoms was hypothesized (*van Hulten et al.*, 2014). Proving the occurrence of Al in





diatoms is key to understanding the mechanism of Al uptake by marine diatoms. Zhou *et al.* investigated Al incorporation into diatoms and proposed the distribution and the content of Al in the organic components of diatoms based on various extraction methods (*Q Liu et al.*, 2019). However, the occurrence of Al in BSi of marine diatoms was still unknown (*Gehlen*

*et al.*, 2002). Al-bearing impurities, which are hardly removed by physical and chemical pre-treatments, influence the detection of the Al incorporated in the BSi, preventing the identification and quantification of the structural Al. Moreover, Al-bearing minerals have been found to form on the surfaces of BSi after diatoms are dead. Due to these problems, the Al detected through elemental analyses such as X-ray fluorescence (XRF), inductively coupled plasma optical emission spectroscopy (ICP-OES) for the DAl in diatom culture medium (*Machill et al.*, 2013), and EDS for marine BSi is unable to

confirm the presence of Al in BSi. Therefore, until now, the lack of direct evidence of Al in BSi has resulted in the existence of structural Al sourced from biological uptake being ambiguous.

*T. weissflogii* shows a high Al tolerance, and its growth is not influenced by Al at such a high initial Al concentration (2.0 μM). These results are in good agreement with the results of some previous reports (*D Liu et al.*, 2019). This high Al tolerance may be due to the biological adaptation of this diatom, which lives in estuary and continental shelf environments

with high environmental Al concentrations (*Zhou et al.*, 2018).

In our previous study, we found that FIB milling allowed to obtain a slice of an internal section of BSi and to remove the external surface, avoiding the disturbance of any impurities in the further detection of the structural elements (*D Liu et al.*, 2019). The freshwater BSi was investigated to reveal Al occurring in the structure of BSi and the high level concentration of Al was detected. However, whether Al is incorporated into the silica structure of marine diatoms or not is still an opening

question. Gehlen *et al.* (2002) and Koning *et al.* (2017) proposed the occurrence of structural Al in marine BSi based on the Al K-edge X-ray absorption near-edge structure (XANES) spectra. However, as declared by Gehlen *et al.* (2002), the detection was impossible to avoid the interference of the Al-bearing phase. Through a combination of FIB and EDS, direct and visible evidence was obtained, illustrating the distribution and quantity of the structural Al in BSi. For the first time, structural Al was observed in BSi of marine diatoms, demonstrating that marine diatoms take up Al and use it to build their

siliceous framework (BSi). Similar to freshwater diatom BSi, Al is homogeneously distributed in marine BSi, but the Al content of marine diatoms (Al/Si atomic ratio of ~0.011) is much lower than that of freshwater diatoms (~0.050) based on the Al/Si atomic ratio (*D Liu et al.*, 2019; *Yuan et al.*, 2019). However, the value is slightly higher than the maximum (0.008) reported in previous studies (*Gehlen et al.*, 2002; *Koning et al.*, 2007). This may be due to the high Al concentration of the culture medium and the preference of the diatoms for Al. It should be noted that the Al/Si atomic ratio of BSi, which was

extracted from living diatoms by removing the organic components of the diatoms, is much larger than the value obtained by FIB-EDS. This result may be due to the disturbance of non-structural Al, which is sourced from the deconstruction of the organic components of the diatoms and is subsequently adsorbed onto the surface of BSi.

Al is incorporated into both the organic components and BSi of diatoms after being taken up through biological behaviors. The Al content of the organic components is higher than that of BSi, indicating that the contribution of the organic

components to the Al occurrence in diatoms is greater than that of the BSi. However, considering the fact that the organic



components start to be degraded by marine bacteria as soon as the diatoms die (*Bidle and Azam*, 1999), the Al in the organic components may return to seawater, which accounts for ~75% of the total Al in the diatoms, and thus only the smaller proportion of the Al that occurs in the BSi structure may be deposited with the BSi. The BSi in sediments constitutes a diatom-driven biological Al sink.

Moreover, structural Al is proposed to have a dissolution-inhibition effect on BSi, i.e., a 25% decrease in BSi solubility when Al substitutes for 1 out of every 70 Si atoms (Al/Si atomic ratio of 0.014) (*Dixit et al.*, 2001). Since the Al/Si atomic ratio of the BSi structure is 0.11, the BSi solubility will be decreased by ~20%, based on the relationship between Al content and BSi dissolution-inhibition. Therefore, this structural Al makes a high contribution to BSi burial. Considering that a huge quantity of carbon settles into the sedimentary layers every year based on BSi exports (*Smetacek*, 1999; *Tréguer et al.*, 2017),

the dissolution-inhibition effect of structural Al on BSi may influence or even control the efficiency of the carbon sequestration of the diatom-driven biological pump.

## 5 Conclusions

In this study, the occurrence of structural Al in BSi of marine diatoms was investigated using a combination of FIB

pretreatment and SEM-EDS analyses. BSi from living *T. weissflogii* diatoms was obtained from culture mediums with an Al concentration of 2.0 μM, indicating that Al is not toxic to *T. weissflogii*. Direct, visible evidence of the presence of structural Al in living BSi was obtained for the first time, and the structural Al was found to have a homogeneous distribution within the silica framework based on the FIB-EDS mapping analysis of BSi slices, the average Al/Si atomic ratio in the structure of the BSi was 0.011, implying that the structural Al makes a high contribution to the dissolution-inhibition of the BSi.

Considering the influence of structural Al on the solubility of diatom biosilica, the occurrence of structural Al in BSi can influence the carbon sequestration efficiency of the diatom-driven biological pump.

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
