# Peer review of "Occurrence of structural aluminium (Al) in marine diatom biological silica: Visible evidence from microscopic analysis"

_Ocean Science, 2021_

## Author Comment (AC1)

| Comments for the Movie and the main text | Responses to the comments |
|---|---|

***Reviewer 1's comment***

This study grew a common cultured centric diatom in low- and high-Al media. Cells from the high-Al media were analyzed for their Al and Si content with focused ion beam scanning SEM and energy dispersive X-ray spectroscopy. This is an interesting and novel dataset that addresses an open question regarding the mechanism through which Al is associated with or incorporated into diatoms in the ocean. The data suggest that Al is contained within the frustule material.

However, I have several concerns with this manuscript. First, few details are provided about the analyses to allow the reader to gauge the quality and consistency of the data. Relatively little of the collected data is actually shown. Additionally, the cells analyzed were grown in media with Al concentrations far exceeding that seen in the ocean, so I think efforts to apply measured Al/Si ratios to the ocean are unwarranted. I expand on these below. While the analytical tools may be well-established, I feel more information is needed about sample preparation and the sequence of analyses for the ocean science reader.

**Authors' general reply:**

We are very grateful for your comments and suggestions, which have been quite helpful in the improvement of our manuscript (Ms). In particular, inspired by some of the comments, we carefully considered the issues mentioned in the comments and improved the Ms accordingly. Please let us know if any more issues need to be clarified or if more revisions need to be made.

In addition to our point-by-point replies (details in the following sections) addressing your concerns, the replies to the major issues and the corresponding revisions are briefly summarized and listed below.

(I) One major concern is that few information of analysis was provided. It is true that this point was not well addressed in the original Ms. In the Method section of the revised Ms, we have supplemented the details, specifically those of sample preparation and analytical processes, including information of diatom culture, composition of artificial sea water, FIB-EDS analysis processes, and EDS precision. We revised the Ms to clarify this issue. Please find details from the replies to **Questions 1, 3, and 4** in the following.

(II) The second major concern is the Al concentration used in our Ms is high. Yes, we used a high concentration of the dissolved Al in our study. The main reason is due to (i) although the Al concentration in open ocean ranges from < 0.1 nM to dozens of nM, the value is up to sever μM in the river mouth and in heavily indus-trialised harbours (*Moran and Moore*, 1988; *Gillmore et al.*, 2016). Our field work also found such the high Al concentration in the marginal sea, such as in Pearl River mouth area where living *T. weissflogii* diatoms were collected. (ii) The *T. weissflogii* diatom has high Al tolerance, which can live in Al-contained diatom culture medium with Al concentration of dozens of μM (*Xie et al.*, 2015; *Vrieling et al.*, 1999), and the high Al concentration can be used. (iii) The incorporated Al can be detected. Accuracy of quantification of EDS used in our study is ≥0.1 wt%, and therefore, to avoid that the content of incorporated Al in BSi is too low to detect, the high Al concentration in the diatom culture medium was selected.

Please keep us informed if any more questions are raised or further discussion is requested. Thank you very much.

***Main comments***

| Comment | Response |
|---|---|
| *1.* The cultures were grown in artificial seawater but no information is given about the composition of this. Supplemental table shows the composition of f/2 media (f/2 ref could be cited and table removed) but nothing about the salt composition of media. | *Yes, the composition of artificial seawater is absent. The corresponding information has been added in the Supporting Information. The supplemental table has been amended, accordingly.* |
| *2.* Additionally, as shown in Fig 3 both 14-day cultures were senescent and not actively growing. In fact, the Al-free culture was actively degrading, although this isn't relevant for the Al measurements, which were done on Al-added cells. Still, the use of senscent cultures raises questions about whether Al is incorporated during normal growth and cell division, or as a feature of cell degradation. | *I do not agree that the diatoms are senescent in 14-day culture, the growing tendency and the final cell concentration of the diatoms are the similar to that described in previous reports (such as (Lang et al., 2013; Vrieling et al., 1999)). We propose that Al-incorporation occurred during the normal growth and cell division because of the distribution of Al in the valves and gridle bands based on EDS results.* |

3. FIB was used to remove the outer layer of the frustule. I am not familiar with this technique and would like to see more detail provided.

3.1 How much material is ablated by the beam? Is it 1/10 of the frustule layer? 1/2? How thin is the slice?

3.2 It isn't clear to me what the 'non-structural elements' are referred to in line 156.

3.3 Is the only difference between the FIB-EDS sample and the BSi sample whether it was milled?

3.4 It isn't clear to me where in fig 4a the 4b and 4c insets are taken from. What part of the frustule? The valve?

3.5 The scale bars in figs 4a, b and c must not all be correct. 4b and 4c look much more zoomed in.

3.6 The EDS spectra used to produce figs 4e-h should be shown, at least one in the paper and the rest in supplemental info.

*Focused ion beam (FIB) thinning, also called FIB milling, is a sample treatment method. The surface of the specimen is etched by ion sputtering as the specimen is irradiated by focused beams of ions (typically $Ga^{+}$), and then surface milling (or thinning) occurs (Marko et al., 2007; Giannuzzi and Stevie, 2005). The purpose of FIB treatment is to expose the internal structure of the specimen for further microscopic characterization. In our study, FIB was used to mill the surface of the frustules to expose the internal for EDS mapping analysis, as displayed in the following figure. In the revised MS, "FIB milling" replaced "FIB thinning" to maintain consistency throughout the entire text. The milling processes are explained in Methods section (lines 122-127) for readers to well understand and the figure is added in the Supporting Information (Fig. S1).*

[Figure]

*Fig. R1 Schematic representation of FIB-milling processes on the BSi particle*

*3.1 Very small part of BSi was obtained after FIB-thinning and maybe the less 1/20 area of the whole frustule was detected. However, the selected regions for detecting Al include those both from valves and gridle bands. The thickness of obtained BSi slices after FIB milling is approximately 80 nm, and the thickness can be controlled by the FIB-milling time. Information has been added in Methods section (lines 125-128).*

*3.2 The expression of 'non-structural elements' meant the adsorbed elements on the external surface of BSi particles. The description of the sentence has been amended, accordingly (lines 158-159).*

*3.3 Yes, the only difference between FIB-EDS sample and BSi sample is that the former has been milled by FIB.*

*3.4 The main figure and the inset in Fig. 4a are the side view and the top view of the frustule, respectively. Fig. 4e and g showed the element distribution of the gridle band and the valve, respectively.*

*3.5 The scales are not correct and we have changed, accordingly. The original figures are shown in Fig. R2 here.*

[Figure]

*Fig. R2 Aluminum element distribution diagram*

*3.6 The EDS spectra have been supplemented in the main text and Supporting Information.*

| | |
|---|---|
| 4. Section 3.3 then presents quantification of Al and Si in the frustules.
4.1 How were spots on the diatoms and BSi differentiated? BSi samples were milled first?
4.2 How far do EDS electrons penetrate into diatom frustule and cell? I'm doubtful the electrons penetrate fully through the cell (for example, see Table 1 in (Twining et al. 2008), allowing accurate measures of Al/Si in the non-frustule parts of the diatom cell.
4.3 How was Al and Si signal quantified? Was sample self-absorption accounted for?
4.4 What support is there the statement on line 174 that Al occurs in the organic component of diatom? The higher Al/Si value for the non-bSi sample could readily be caused by the lower Si concentration of this fraction. This would be clarified by presenting the actual Al and Si concentration data and not just the ratios. | *4.1 BSi was obtained by extraction from the entire diatoms (details see Section 2.2) without FIB treatments. We selected the spots in diatoms and their BSi along the diagonal line and at least 10 spots are analyzed for one diatom or BSi particle (see the following schematic representation, Fig. R3). The position of data collection included those in valves and gridle bands. The information has been added in the Supporting Information.*

[Figure]

*Fig. R3. Schematic representation of the selection of EDS spots (red spots are the position of EDS data collection)*
*4.2 Yes, the Conventional electrons penetrate fully through the cell. Therefore, we chose FIB to prepare BSi samples, remove other parts of diatom cells and accurately detect BSi.*
*4.3 The EDS spot analysis used to determine the Al/Si atomic ratios was based on more than 300 spots on 35 diatom cells and their BSi samples.The raw EDS data of the atomic percentage (at%) of the elements were obtained. In addition, they were transformed to the relative atomic concentration of the elements (to Si atomic concentration), as a normalization that excludes the influence of nonindigenous elements.Many previous publications (e.g., Gehlen, et al., 2002; Ingall et al., 2013; Michalopoulos, 2000) also used the relative concentration (e.g., Al/Si molar ratio) instead of absolute concentration.*
*4.4 Yes, This statement is not accurate enough. We agree that the atomic percentage of Si is lower in the entire diatoms than in BSi, so is the atomic percentage of Al, due to the existence of carbon and other elements. Some studies have shown that aluminum binds to protein in diatoms, most of it is loose particle aluminum (Liu et al., 2019).* |
| 5. The paper presents an average Al/Si ratio of ~0.011 measured for the cultured diatoms and equates this to the Al content of marine diatoms in the ocean (line 211). I think the use of 2 μM Al solution (about 100-fold higher than Al in ocean surface waters) makes it problematic to conclude that the resulting Al content of the diatom opal is representative of ocean diatoms. This needs to be addressed in the manuscript. | *In this manuscript, we aimed to reveal the occurrence of structural Al in BSi, and based on the simulation experiments we cultured diatoms with high Al-tolerance in the culture medium possessing a relatively high Al-concentration. Indeed, in the open ocean, the dissolved Al concentration is much less than the value (2 uM) and therefore, it can not be used to evaluate the content of Al-corporation in the BSi of living diatoms. We added some interpretations in the text for addressing it.* |

| *Other specific comments* | |
|---|---|
| 1. Line 187: refers to Zhou et al. but cites Liu et al. Please cite the correct reference. | *Yes, the references have been corrected.* |
| 2. *Line 192: please provide a reference for this statement.* | *Yes, new references have been added in the main text.* |
| 3. Lines 197-200: I find the short sections on the Al tolerance of the diatom out-of-place in this paper. It is not a paper about Al tolerance, and this aspect is not treated rigorously (for example, there is no discussion or consideration of Al speciation, and no other species were tested); I suggest removing these sections. | *The section has been removed, accordingly.* |
| 4. Line 210: What is the evidence for homogeneous distribution of Al in BSi? Figs 4e and g? Fig 4g doesn't show particularly homogeneous Al distribution, in my opinion, nor does the image provide a scale (it appears to be binary). This statement would be strengthened by showing more data. | *Yes, the word of "homogeneous" used in the manuscript is not accurate. We meant the similar distribution characteristics with Si, that is, the higher the Si concentration is, the higher the Al concentration is in BSi. The text has been amended, accordingly.* |
| 5. Line 211: Given the small amount of data provided and the use of 2 µM Al in the culture media (approx. 100-fold over ambient concentrations), I do not feel it is justified to present the measured Al/Si ratio is representative of marine diatoms. Additionally, diatoms vary significantly in their level of silicification (Baines et al. 2010), which could well also affect this ratio. | *Yes, the simulation experiment aims to characterize the Al occurrence and thus the Al concentration is much higher than that in seawater of open ocean for the purpose of easily detecting, as mentioned above. Therefore, the result of Al content in BSi can be not representative of the value in marine diatoms. Indeed, the Al/Si ratios are various in different marine diatoms, as reported in previous studies, and the maximum is nearly 0.008 (Gehlen et al., 2002; Koning et al., 2007), which may be attributed to the various levels of silicification. The description of the text has been amended.* |
| 6. L218: what is meant by 'biological behaviors'? What evidence is provided of this? | *"Biological behaviors" is the speculation. It results from the studies in this and other reports that Al was incorporated into the diatom cell. This Al uptake of diatoms is proposed to be attributed to the biological behaviors (Gélabert et al., 2004; Van Bennekom et al., 1989; Liu et al., 2019), and we used the expression for explain the Al incorporation in BSi. However, we have no direct evidence for supporting it. Therefore, we have deleted the phrase.* |
| 7. L219: what evidence is provided that Al content of organic components is higher than that of BSi? | *We can not provide the evidence of Al content in organic components higher than in BSi, due to no data of the Al in organic components. Moreover, this study focuses on the evaluation of incorporated Al in BSi, and thus the expression has been deleted in the text, accordingly.* |
| 8. L222: how is it calculated that organic components account for 75% of Al in diatoms? | *Text has been amended, accordingly.* |
| 9. L224: This statement about sediments as an Al sink would be strengthened by a comparison of Al/Si in sediments and diatoms. How much sedimentary Al might be attributed to diatoms? | *We believe that the contribution of the Al in sedimentary BSi to the Al in the sediments is very low. This is because abundant clay minerals in the sediments contribute plenty of Al. The content of BSi is very low as well as the incorporated Al. Therefore, we just take the BSi as a biological Al sink and we can not provide the specific data for the contribution only based on this research in the lab at this stage.* |

**References**

Gehlen, M., Beck, L., Calas, G., Flank, A. M., Van Bennekom, A. J., and Van Beusekom, J. E. E.: Unraveling the atomic structure of biogenic silica: Evidence of the structural association of Al and Si in diatom frustules, Geochimica Et Cosmochimica Acta, 66, 1601-1609, https://doi.org/10.1016/s0016-7037(01)00877-8, 2002.

Gélabert, A., Pokrovsky, O. S., Schott, J., Boudou, A., Feurtet-Mazel, A., Mielczarski, J., Mielczarski, E., Mesmer-Dudons, N., and Spalla, O.: Study of diatoms/aqueous solution interface. I. Acid-base equilibria and spectroscopic observation of freshwater and marine species1 1Associate editor: J. B. Fein, Geochimica et Cosmochimica Acta, 68, 4039-4058, https://doi.org/10.1016/j.gca.2004.01.011, 2004.

Giannuzzi, L. A. and Stevie, F. A.: Introduction to Focused Ion Beams: Instrumentation, Theory, Techniques and Practice, Springer, https://doi.org/10.1007/0-387-23313-X_2, 2005.

Gillmore, M. L., Golding, L. A., Angel, B. M., Adams, M. S., and Jolley, D. F.: Toxicity of dissolved and precipitated aluminium to marine diatoms, Aquatic Toxicology, 174, 82-91, https://doi.org/10.1016/j.aquatox.2016.02.004, 2016.

Ingall, E. D., Diaz, J. M., Longo, A. F., Oakes, M., Finney, L., Vogt, S., Lai, B., Yager, P. L., Twining, B. S., and Brandes, J. A.: Role of biogenic silica in the removal of iron from the Antarctic seas, Nature Communications, 4, 1981, https://doi.org/10.1038/ncomms2981, 2013.

Koning, E., Gehlen, M., Flank, A. M., Calas, G., and Epping, E.: Rapid post-mortem incorporation of aluminum in diatom frustules: Evidence from chemical and structural analyses, Marine Chemistry, 106, 208-222, https://doi.org/10.1016/j.marchem.2006.06.009, 2007.Lang, Y., Monte, F. d., Rodriguez, B. J., Dockery, P., Finn, D. P., and Pandit, A.: Integration of TiO2 into the diatom Thalassiosira weissflogii during frustule synthesis, Scientific Reports, 3, 3205, https://doi.org/10.1038/srep03205, 2013.

Lang, Y., Monte, F. d., Rodriguez, B. J., Dockery, P., Finn, D. P., and Pandit, A.: Integration of TiO2 into the diatom Thalassiosira weissflogii during frustule synthesis, Scientific Reports, 3, 3205, https://doi.org/10.1038/srep03205, 2013.

Liu, Q., Zhou, L., Liu, F., Fortin, C., Tan, Y., Huang, L., and Campbell, P. G. C.: Uptake and subcellular distribution of aluminum in a marine diatom, Ecotoxicology and Environmental Safety, 169, 85-92, https://doi.org/10.1016/j.ecoenv.2018.10.095, 2019.

Marko, M., Hsieh, C., Schalek, R., Frank, J., and Mannella, C.: Focused-ion-beam thinning of frozen-hydrated biological specimens for cryo-electron microscopy, Nature Methods, 4, 215-217, https://doi.org/10.1038/nmeth1014, 2007.

Michalopoulos, P., Aller, R. C., and Reeder, R. J.: Conversion of diatoms to clays during early diagenesis in tropical, continental shelf muds, Geology, 28, 1095-1098, https://doi.org/10.1130/0091-7613(2000)28<1095:codtcd>2.0.co;2, 2000.

Moran, S. B. and Moore, R. M.: Evidence from mesocosm studies for biological removal of dissolved aluminum from sea-water, Nature, 335, 706-708, https://doi.org/10.1038/335706a0, 1988.Rich, H. W. and Morel, F. M. M.: Availability of well-defined iron colloids to the marine diatom Thalassiosira weissflogii, Limnology and Oceanography, 35, 652-662, https://doi.org/10.4319/lo.1990.35.3.0652, 1990.

van Bennekom, A. J., Fred Jansen, J. H., van der Gaast, S. J., van Iperen, J. M., and Pieters, J.: Aluminium-rich opal: an intermediate in the preservation of biogenic silica in the Zaire (Congo) deep-sea fan, Deep Sea Research Part A. Oceanographic Research Papers, 36, 173-190, https://doi.org/10.1016/0198-0149(89)90132-5, 1989.

Vrieling, E., Poort, L., Beelen, T., and Gieskes, W.: Growth and silica content of the diatomsThalassiosira weissflogiiandNavicula salinarumat different salinities and enrichments with aluminium, European Journal of Phycology, 34, 307-316, https://doi.org/10.1080/09670269910001736362, 1999.

Xie, J., Bai, X., Lavoie, M., Lu, H., Fan, X., Pan, X., Fu, Z., and Qian, H.: Analysis of the Proteome of the Marine Diatom Phaeodactylum tricornutum Exposed to Aluminum Providing Insights into Aluminum Toxicity Mechanisms, Environmental Science & Technology, 49, 11182-11190, https://doi.org/10.1021/acs.est.5b03272, 2015.

---

## Author Comment (AC2)

| Comments for the Movie and the main text | Responses to the comments |
|---|---|

The present article deals with the occurrence of Al in diatoms frustules. Data are presented that show that Al is also found in the inner parts of diatoms frustules. This underlines that Al is taken up and incorporated in the frustules and excludes that Al is from other sources like detrital aluminium silicates. These data are a valuable addition to our knowledge of Al in BSi and should be published.

However, the authors state that the "occurrence mechanism of Al in marine diatoms is unclear. In particular, whether or not Al is incorporated in the structure of BSi." This is in my view not correct: Gehlen et al. (2002, cited by the authors) already gave evidence that Al is incorporated in frustules. They also showed that Al is present in the Si structure of cultured diatoms exclusively is in a four-fold coordination. Additional work by Beck et al. (2002, not cited by the authors) show that Ca is present in proportion to $Al^{3+}$ further supporting the structural incorporation of Al in the $SiO_2$ network of frustules.

In my view, this work has to be acknowledged and the research question of the paper should be adapted to this.

**Authors' general reply:**

We are very grateful for your comments and recognition of our work, which have been quite helpful in the improvement of our manuscript. Prof. Beck and Prof. Gehlen proposed Al occurrence in BSi based on the XANES analysis, showing tetrahedral coordination of Al in BSi, and we agree that the Al was attributed to the incorporated Al in BSi. However, the visible, evidence lacks and the XAS analysis can not showing the content and distribution of structural Al. Moreover, the Al-detection by XAS is difficult in avoiding the interference of adsorbed Al and Al-bearing minerals from the sediments which may contain some minerals with tetrahedral coordinated Al, such as illite and mica, as described in the paper (*Despite our best efforts to isolate clean diatom frustules, a contamination by adhering clay particles cannot be excluded at this stage. Gehlen et al, 2002*). Therefore, our study provided a visible evidence of Al in BSi, showed more information about incorporated Al in BSi, and proposed an effective method to detect the BSi by avoiding any interference of adsorbed Al and Al-contained minerals. Considering the contribution of Beck and Gehlen, we changed the expression, accordingly.

The replies to the major issues and the corresponding revisions are briefly summarized and listed below. Please keep us informed if any more questions are raised or further discussion is requested. Thank you very much.

*Main comments*

| | |
|---|---|
| 1. L20: "occurrence mechanism" should be replaced by a more clearer wording | *The text has been amended, accordingly.* |
| 2. L37: Given the global perspective of the paper, the range of Al concentrations can be specified ranging from below 1 nM in the Southern Ocean (e.g., van Beusekom et al., 1997) to well above 100 nM in the Mediterranean (e.g., Chou & Wollast, 1997). | *The text has been amended, accordingly.* |
| 3. L55: Here, the work by Gehlen et al. (2002) and Beck et al. (2002) showing that Al is a structural part of the BSi should be cited. Note that Gehlen et al. acknowledge that Al bearing minerals may interfere when analyzing field material, but they also report on cultured diatoms grown in mediums well below the concentrations used in the present study. | *The references have been added.* |
| 4. L65 ff: These statements need to be precised regarding the work by Beck et al. and Gehlen et al. who actually showed (not proposed) the coordination of Al in BSi (and the compensation of charge by inclusion of Ca). | *Yes, we have added the corresponding information about the research findings of Beck et al. and Gehlen et al in the text.* |

| | |
|---|---|
| **5.** L74-75: This is an unclear sentence that needs rewording. | *The text has been amended, accordingly.* |
| **6.** L90. Figure 1: misspelling of "isolate" | *Yes, we have corrected the wrong words in the picture.* |
| **7.** L94 Light intensity of 100 µmol photons…. (isn't the unit in µmol?) | *Yes, we have corrected the wrong unit.* |
| **8.** L95: F/2 was used. This contains Fe-EDTA. A large part of the Fe may be released and precipitate, impacting the Al concentrations. Given the very high concentrations used (2000 nM (10-fold of max concentrations in the ocean) precipitates may occur. Please comment on this. | *Yes, the F/2 medium contains Fe-EDTA, which is of bioavailability for diatoms (Rich and Morel, 1990). Alcan hardly form complexes with EDTA in such alkaline seawater (Zhou et al., 2016; Liu et al., 2019; Zhou et al., 2021). In this scenario, EDTA may cause little influence on the Al uptake of diatoms. Moreover, we checked the F/2 with 2 µM Al before diatoms were added and Al-bearing precipitates were not found.* |
| **9.** L101: Please indicate the Al concentration of the medium without Al. These concentrations can be substantial. | *We measured the concentration of dissolved Al before Al was added by Ultraviolet–visible Spectrophotometer and the results showed the Al concentration of several nM.* |
| **10.** L130: Are the authors referring to their own studies? This should be cited. | *The references have been added, and the text has been amended, accordingly.* |
| **12.** L225: Here, the work by van Cappellen et al. on the effect of Al on the dissolution rate would be appropriate. | *Yes, the reference has been added to the main text.* |
| **13.** L236: The conclusion that 2000 nM Al is not toxic should be discussed in the light of the possibility that part of the Al is co-precipitated with Fe from the F/2 Trace metal mix. Furthermore, it has to be clarified whether EDTA is able to build a complex with EDTA. This is also relevant for the cleaning procedure where EDTA is used to desorb impurities. | *Al is confirmed not to form complexes significantly with EDTA in alkaline seawater. In addition, thus the high Al-tolerant of diatoms was not attributed to the influence of EDTA (Canterford, 1979). In addition, many previous studies also indicated the diatom has high Al-tolerant (Xie et al., 2015; Gillmore et al., 2016). As suggested by Reviewer 1, we deleted the section of Al-tolerant.* |
| **14.** L237: The statement that structural Al in BSi is demonstrated for the first time is in my view not valid, as Gehlen et al. and Beck et al. this already demonstrated. The observation that Al is present throughout the BSi is a good addition to our knowledge of Al in BSi. | *The text has been changed, accordingly.* |

**References**

Canterford, G.: Effect of EDTA on Growth of the Marine Diatom Ditylum brightwellii (West) Grunow, Marine and Freshwater Research, 30, 765-772, https://doi.org/10.1071/MF9790765, 1979.

Gehlen, M., Beck, L., Calas, G., Flank, A. M., Van Bennekom, A. J., and Van Beusekom, J. E. E.: Unraveling the atomic structure of biogenic silica: Evidence of the structural association of Al and Si in diatom frustules, Geochimica Et Cosmochimica Acta, 66, 1601-1609, https://doi.org/10.1016/s0016-7037(01)00877-8, 2002.

Gillmore, M. L., Golding, L. A., Angel, B. M., Adams, M. S., and Jolley, D. F.: Toxicity of dissolved and precipitated aluminium to marine diatoms, Aquatic Toxicology, 174, 82-91, https://doi.org/10.1016/j.aquatox.2016.02.004, 2016.

Liu, Q., Zhou, L., Liu, F., Fortin, C., Tan, Y., Huang, L., and Campbell, P. G. C.: Uptake and subcellular distribution of aluminum in a marine diatom, Ecotoxicology and Environmental Safety, 169, 85-92, https://doi.org/10.1016/j.ecoenv.2018.10.095, 2019.

Rich, H. W. and Morel, F. M. M.: Availability of well-defined iron colloids to the marine diatom Thalassiosira weissflogii, Limnology and Oceanography, 35, 652-662, https://doi.org/10.4319/lo.1990.35.3.0652, 1990.

Xie, J., Bai, X., Lavoie, M., Lu, H., Fan, X., Pan, X., Fu, Z., and Qian, H.: Analysis of the Proteome of the Marine Diatom Phaeodactylum tricornutum Exposed to Aluminum Providing Insights into Aluminum Toxicity Mechanisms, Environmental Science & Technology, 49, 11182-11190, https://doi.org/10.1021/acs.est.5b03272, 2015.

Zhou, L., Tan, Y., Huang, L., and Wang, W.-X.: Enhanced utilization of organic phosphorus in a marine diatom Thalassiosira weissflogii: A possible mechanism for aluminum effect under P limitation, Journal of Experimental Marine Biology and Ecology, 478, 77-85, https://doi.org/10.1016/j.jembe.2016.02.009, 2016.

Zhou, L., Liu, F., Liu, Q., Fortin, C., Tan, Y., Huang, L., and Campbell, P. G. C.: Aluminum increases net carbon fixation by marine diatoms and decreases their decomposition: Evidence for the iron–aluminum hypothesis, Limnology and Oceanography, 66, 2712-2727, https://doi.org/10.1002/lno.11784, 2021.

---

## Referee Report (RR1)

Review of
Occurrence of structural aluminium (Al) in marine diatom biological silica: Visible evidence from microscopic analysis by Tian et al.,
The revised version has clearly improved compared to the first version.
However, a few points remain.

Abstract:
Line 25/26: This is not the first time, as Gehlen et al and Beck et al already gave evidence for structural Al in BSi. The main point in this study is in my view, that it was shown that Al is found throughout the BSi, suggesting a direct link between Al incorporation and Si incorporation.

Introduction:
L38. I suggest to split the sentence into two sentences.
L69: Misspelling of Gehlen
L72: This was not speculated but shown
L73: I suggest transfer instead of migration

M&M
L96: Composition is (not are)
L 103: the initial Al concentration is not mentioned (but indicated in the reply) and should be mentioned here.

Results
L198/L211: Here, the structural evidence by Gehlen et al and Beck et al. should be discussed in more detail and their evidence of structural Al based on the coordination of Al and Si should be mentioned in the text. Also, it should be pointed out that in the measurements from cultures, naturally occurring Al/si containing minerals do not play a role.
L230: the Al/Si should be 0.011, not 0.11? Also mention, that the ratio is based on the FIB treated BSi

---

## Author Response (AR2)

**Response to comments-OS-2021-78**

*Editor's Comments:*

I have sent your revised manuscript to one of the reviewers, who finds it improved, but demands that the work of Gehlen and Beck be discussed in more detail. You need to emphasize what is new in your study, in relation to the previous work on the same topic. I hope that you can consider the reviewer's remarks in a revised version of your manuscript.

**Authors' general reply:**

We are very grateful for your comments. We have carefully checked and amended the manuscript accordingly. In addition to our point-by-point replies (details in the following sections) addressing your concerns, the replies are briefly summarized and listed below.

Please let us know if any more issues need to be clarified or if more revisions need to be made.

Thank you very much.

*Reviewer's comments*

| Comments | Responses to the comments |
|---|---|
| *1.* Line 25/26: This is not the first time, as Gehlen et al and Beck et al already gave evidence for structural Al in BSi. The main point in this study is in my view, that it was shown that Al is found throughout the BSi, suggesting a direct link between Al incorporation and Si incorporation. | *Yes, the text has been changed, accordingly.* |
| *2.* L38. I suggest to split the sentence into two sentences. | *Yes, the text has been changed, accordingly.* |
| *3.* L69: Misspelling of Gehlen | *Yes, we have corrected the wrong spelling.* |
| *4.* L72: This was not speculated but shown | *Yes, the text has been changed, accordingly.* |
| *5.* L73: I suggest transfer instead of migration | *Yes, the text has been changed, accordingly.* |
| *6.* L96: Composition is (not are) | *Yes, we have corrected the wrong unit.* |
| *7.* L103: the initial Al concentration is not mentioned (but indicated in the reply) and should be mentioned here. | *Yes, the data were added into the main text.* |

| | |
|---|---|
| **8.** L198/L211: Here, the structural evidence by Gehlen et al and Beck et al. should be discussed in more detail and their evidence of structural Al based on the coordination of Al and Si should be mentioned in the text. Also, it should be pointed out that in the measurements from cultures, naturally occurring Al/Si containing minerals do not play a role. | *Yes, we have added the corresponding information about the structural Al in BSi reported by Beck et al. and Gehlen et al in the main text.* |
| **9.** L230: the Al/Si should be 0.011, not 0.11? Also mention, that the ratio is based on the FIB treated BSi | *Yes, the data have been corrected, and the corresponding information has been added.* |